# Liquid Biopsy Biomarkers for Immunotherapy in Non-Small Cell Lung Carcinoma: Lessons Learned and the Road Ahead

**DOI:** 10.3390/jpm11100971

**Published:** 2021-09-28

**Authors:** Jesus Hita-Millan, Angel Carracedo, Ceres Fernandez-Rozadilla

**Affiliations:** 1Grupo de Medicina Xenómica, Universidade de Santiago de Compostela (USC), 15706 Santiago de Compostela, Spain; jess.hita@rai.usc.es (J.H.-M.); angel.carracedo@usc.es (A.C.); 2Instituto de Investigación Sanitaria de Santiago (IDIS), 15706 Santiago de Compostela, Spain; 3Fundación Pública Galega de Medicina Xenómica, SERGAS, 15706 Santiago de Compostela, Spain; 4Consorcio Centro de Investigación Biomédica en Red de Enfermedades Raras Fundación CIBERER, 28029 Madrid, Spain

**Keywords:** NSCLC, biomarkers, liquid biopsy, cancer treatment, immune checkpoint inhibitors, immunotherapy, personalised medicine, CTCs, extracellular vesicles, ctDNA

## Abstract

Over the recent years, advances in the development of anti-cancer treatments, particularly the implementation of ICIs (immune checkpoint inhibitors), have resulted in increased survival rates in NSCLC (non-small cell lung cancer) patients. However, a significant proportion of patients does not seem respond to immunotherapy, and some individuals even develop secondary resistance to treatment. Therefore, it is imperative to correctly identify the patients that will benefit from ICI therapy in order to tailor therapeutic options in an individualised setting, ultimately benefitting both the patient and the health system. Many different biomarkers have been explored to correctly stratify patients and predict response to immunotherapy, but liquid biopsy approaches have recently arisen as an interesting opportunity to predict and monitor treatment response due to their logistic accessibility. This review summarises the current data and efforts in the field of ICI response biomarkers in NSCLC patients and highlights advantages and limitations as we discuss the road to clinical implementation.

## 1. Introduction

Lung cancer is the second most commonly diagnosed cancer globally and the number one cause of cancer-related death in the world (18% of all cancer deaths), with estimates of deaths surpassing 1,796,000 cases worldwide in 2020 [1,2]. Lung cancer can be divided into three different categories: non-small cell lung cancer (NSCLC; 80% of cases), small cell lung cancer (SCLC; 15–20% of cases), and other lung carcinoid tumours [2]. Standard management of lung cancer depends on disease stage, with typical strategies relying on tumour resection followed by adjuvant chemotherapy and/or radiotherapy. In the case of advanced NSCLC, 4–6 cycles of cisplatin or carboplatin has been the standard chemotherapy treatment [3]. In the past few years, the contemplation of molecular features, particularly using next-generation sequencing (NGS), has allowed the implementation of therapies targeted to specific actionable alterations (e.g., *ROS1*, *EGFR*, *ALK*, *MET*, *KRAS*, *ERBB2*, *RET*), albeit in a small proportion of cases [4,5,6,7,8]. Nevertheless, overall survival (OS) rates remain low, and many patients develop secondary resistance during treatment [9,10].

In the past few years, immunotherapy has also become available for advanced-stage NSCLC patients in addition to standard chemo/radiotherapy if there is detectable expression of programmed death ligand-1 (PD-L1) [11]. Immunotherapy, administered as first- or second-line treatment, produces a considerable improvement in NSCLC patient survival. However, determining eligibility and response to immunotherapy is key to ensure its effective use in NSCLC, since 70–80% of NSCLC patients that receive it do not respond to treatment [12].

In this review, we summarise the current state of biomarkers that can be used to monitor response to immunotherapy in NSCLC patients, with a particular emphasis on the liquid biopsy methods currently available and the potential for novel biomarker identification. These liquid biopsy approaches may be particularly useful for providing personalised dynamic biomarkers in the near future that can be used in a precision oncology setting.

## 2. The PD-L1/PD-1 Axis

The PD-L1/PD-1 axis is a self-tolerance mechanism that protects cells from being targeted for immune destruction during an inflammatory response. Activated cytotoxic T lymphocytes search for foreign peptides bound to MHC molecules on the surface of cells, binding to them to activate the inflammatory cascade. This binding drives the production of inflammatory cytokines that trigger the expression of *PD-L1* (encoded by the *CD274* gene) on the surface of normal cells, and PD-L1 subsequently interacts with programmed death protein-1 (PD-1) on the surface of T cells, leading to immune tolerance. This renders the T cell unable to carry out immune destruction of the normal cells (Figure 1A) [13]. Although tumour cells are normally targeted for immune destruction by cytotoxic T cells (Figure 1B), many cancer cells have evolved to exploit this system to avoid immune destruction in the same manner. As first observed in melanoma, some tumour cells can overexpress *PD-L1*, increasing the interaction between PD-L1 and PD-1 and thus inactivating T lymphocytes that attempt to trigger an immune response against the tumour (Figure 1C). This achieves both the immune evasion of the tumour cells as well as creates a more aggressive tumour microenvironment by promoting the production of pro-inflammatory cytokines and causing tumour-infiltrating lymphocyte (TIL) and cytotoxic T-cell (CD8+) exhaustion, which thwarts any attempts of immune targeting by the local T cells [14].

## 3. Immune Checkpoint Inhibitors and Immunotherapy 

In the later 20th century, blockage of the interaction between PD-L1 and PD-1 was proposed as a possible therapeutic path to combat tumour growth, and studies using monoclonal antibodies targeting immune checkpoints (such as PD-L1/PD-1) started to proliferate (Figure 1D). These antibodies are referred to as immune checkpoint inhibitors (ICIs). It was clear from the early studies with mice xenografts that treatment with ICIs produced recovery of effector T-cell functions and a decrease in tumour size [15,16]. 

In 2015, the Food and Drug Administration (FDA) approved the use of the first anti-PD-1 monoclonal antibody (Nivolumab) as second-line treatment of NSCLC after the results of the clinical trials *CheckMate 017* and *CheckMate 057* showed an improved response over standard chemotherapy in NSCLC patients [17]. Moreover, *CheckMate 017* showed an improved OS in NSCLC patients treated with Nivolumab compared to standard treatment with Docetaxel (median OS: 9.2 months vs. 6.0 months, respectively) [18]. In 2016, the FDA also approved the use of Pembrolizumab (anti-PD-1) as first- or second-line monotherapy treatment in NSCLC patients and Atezolizumab (anti-PD-L1) for patients with disease progression during or following platinum-based chemotherapy [19,20,21,22]. Multiple ICIs have since been developed and approved by the FDA and other government agencies (Table 1).

## 4. ICI Response Biomarkers

In essence, the purpose of immunotherapy is to re-awaken the immune system to allow the targeting of cancer cells for immune destruction, with ICIs that block PD-1 and PD-L1 being used to hinder the immune tolerance mechanism triggered by the PD-1/PD-L1 interaction. However, although immunotherapy works well for a subset of cancer patients, a significant portion of individuals do not benefit from ICI treatment, and the exact mechanisms that prevent these patients from responding have not yet been elucidated. In NSCLC for example, patients had an objective response rate (ORR) of 30.2% in first line, while the ORR in second-line patients was 20.1% [12]. 

Since a considerable proportion of patients does not benefit from immunotherapy, it is necessary to identify biomarkers that accurately predict the subset of responders, thereby allowing either rapid and effective treatment or early evaluation of alternative therapeutic strategies. The current guidelines of immunotherapeutic treatment base eligibility of patients for PD-1/PD-L1 inhibitors mainly on PD-L1 levels on the primary tumour or, as of recently, on tumour-mutation burden (TMB) levels [24,25,26]. However, these biomarkers are not optimally accurate, and as we will discuss below, it is necessary to explore alternatives that provide a more effective predictive stratification of patient response.

### 4.1. PD-L1 Expression as a Response Biomarker

Soon after the first ICI trials, it was observed that PD-L1 levels were correlated with treatment response. According to the *KEYNOTE-042* trial, the group of NSCLC patients treated with Pembrolizumab that showed ≥50% cells with PD-L1 expression in their primary tumour experienced a greater decrease in tumour mass compared to the group with PD-L1 positivity between 1% and 49%, which in turn also performed better than the group with less than 1% PD-L1+ cells [27,28]. This correlation between a higher PD-L1 and better response to ICI immunotherapy resulted in the establishment of *PD-L1* expression evaluation prior to treatment as a means to identify patients that could benefit from immunotherapy. Although various assays are available that comprise different scoring systems (*Ventana SP263*, *Ventana SP142*, *Dako28-8*, and *Dako 22C3*), they are all based on immunohistochemical staining of tumour cells for PD-L1 and the calculation of the percentage of PD-L1 cell positivity [29]. These tests have become standard practice and have led to the inclusion of PD-L1 expression as a determining factor of eligibility for NSCLC patients to undergo ICI treatment. According to the European Society of Medical Oncology (ESMO) and the National Comprehensive Cancer Network (NCCN) guidelines, PD-1/PD-L1 inhibitors are recommended as first-line monotherapy treatment only for NSCLC patients with ≥50% PD-L1 expression and no other actionable molecular markers, while ICI treatment for NSCLC patients with 1–49% PD-L1 expression is only recommended as first-line therapy if in conjunction with a platinum-based chemotherapeutic agent and Pemetrexed or as second-line monotherapy treatment [24,25]. Although immunotherapy is not recommended in patients showing <1% PD-L1 expression in any of the guidelines, it can be considered when no other treatment options are available (compassionate use).

PD-L1 expression has rapidly become the most accepted predictor of response to anti-PD1/PD-L1 ICIs, but it is far from being a perfect biomarker. In *CheckMate 017* and *CheckMate 057*, treatment with Nivolumab showed improved response rates and prognosis when PD-L1 was assessed as detectable (>1%) in non-squamous NSCLC but not in the squamous subtype [18,28]. The *ATLANTIC* study showed that 6.6% of NSCLC patients that presented an OR after treatment with Durvalumab showed no discernible PD-L1 expression [30]. This shows that the use of PD-L1 expression alone may prevent patients that may benefit from ICI treatment from receiving it. Therefore, it is imperative to identify more accurate ICI response biomarkers to tailor the resources to those most likely to benefit. 

### 4.2. Tumour Mutation Burden

TMB is defined as the density of non-synonymous somatic mutations in a tumour genome, expressed as number of mutations per megabase (mut/Mb) [31]. It has sparked interest as an ICI response biomarker because a high TMB harbours an increased chance of developing neoantigens, and hence, it is assumed that it will correlate with a reactivation of T cells and a better response [32,33].

Therefore, the relationship between TMB and response to ICIs has been explored over the years, and it is established that patients harbouring >10 mut/Mb have increased response rates and a higher progression-free survival (PFS) in certain cancer types, like bladder cancer, melanoma, and NSCLC [34,35]. For NSCLC in particular, different studies have shown that TMB in the primary tumour tends to be higher in men than in women and is also positively correlated with tobacco consumption, *KRAS*, and *BRAF* mutated tumours, whereas the presence of *ROS1* and *ALK* rearrangements is linked to lower TMB [36,37,38,39,40].

Technically, whole exome sequencing (WES) was the initially proposed strategy for measuring TMB [41]. However, WES is costly, time consuming, and demands considerable data storage for large numbers of samples. Hence, smaller NGS panels have been developed and validated to perform TMB calculations on a clinical setting basis. Although many panels validated for TMB calculation are currently available, only two panels have been approved by the FDA: the Foundation One CDx panel, which includes 324 cancer genes (0.8 Mbp), and the MSK-IMPACT, covering 468 cancer-related genes (1.5 Mb) [42,43,44,45]. The FDA even adjusted immunotherapy guidelines by recommending the use of Pembrolizumab in tumour mutational burden-high (TMB-H) solid tumours in patients that have progressed and have no other treatment options even if PD-L1 levels in the primary tumour are <1% [26].

Unfortunately, TMB still remains as a very imperfect biomarker, as high TMB does not guarantee a response to ICIs, and low TMB does not exclude the possibility of response to immunotherapy. This was shown in trial *CheckMate 026* when a group of 158 NSLC patients treated with Nivolumab were assessed for percentage of PD-L1-positive cells and TMB [46]. A higher ORR was detected in patients that had both high *PD-L1* expression and high TMB (75%) compared to patients with only one of these factors (32% in only TMB-H patients and 34% in PD-L1-high patients) or patients with low *PD-L1* expression and low TMB (18%). This suggests that a significant percentage of patients with high TMB do not benefit from ICIs, while some patients with low TMB may actually do. Moreover, TMB has failed to predict response in other cancer types, such as breast, prostate, and head and neck cancer [47]. This has raised concerns over the use of immunotherapy in any TMB-H cancer, arguing that TMB should be used as a biomarker of response only in cancer types that show a correlation between neoantigen load and T_CD8+_ cell tumour infiltration and not in all solid tumours, as recommended by the FDA. In addition to this, the pan-cancer threshold of ≥10 mut/Mb recommended by the FDA has been challenged by numerous studies, showing that different optimal thresholds should be adjusted according to the type of cancer observed and panel used [47,48,49]. This need for a consensus demands further studies to establish TMB as a valid ICI response biomarker.

### 4.3. Other Tumour-Based Markers

Cancer cells are surrounded by several adjacent cell types in what is known as the tumour microenvironment (TME). The TME includes a variety of endothelial cells, stromal fibroblasts, immune cells, signalling molecules, and the extracellular matrix, and its interplay with cancer cells is relevant to processes such as neo-angiogenesis, immune evasion, and tissue remodelling [50]. All these players are thus susceptible to be used in r biomarker studies. 

One of the most relevant aspects of the TME in relation to immunotherapy is the determination of adaptive immune resistance. In early studies, the presence of TILs in tumours was associated to an increased *PD-L1* tumour expression mediated by interferon gamma (IFN-γ). IFN-γ is a cytokine (enconded by the *IFNG* gene) produced by cytotoxic T cells and natural killer (NK) cells during immune response. When released, it recruits more T cells to the tumour site to boost the antiproliferative response [51]. However, IFN-γ also triggers the up-regulation of *PD-L1* expression in adjacent tumour cells to avoid immune recognition and destruction by T cells in a process called adaptive immune resistance [52]. Because of this, blockage of the PD-1/PD-L1 axis may restart the anti-tumour activity of IFN-γ in responders, and observation of *PD-L1* expressed in the tumour together with the presence of TILs was therefore proposed to determine whether a tumour is exhibiting immune evasion and could benefit from immunotherapy [53].

Recently, stratification of the TME depending on the PD-L1 status and presence of TILs has been proposed resulting in four types of TME [54]: type I, with PD-L1+ tumour and high presence of TILs driving adaptive immune resistance; type II, with PD-L1- tumour and low presence TILs; type III, with PD-L1+ tumour and low presence of TILs; and type IV, with PD-L1- but high presence of TILs promoting immune tolerance of the tumour. For NSCLC, type II tumours are the most frequent [55]. 

Overall, it is not just TIL presence within the tumour stroma that matters but high counts of infiltrating NK cells, and dendritic cells have been associated with a better general response and prognosis and a higher OS after treatment with ICIs [56,57,58]. Thrombocytes are also considered as both biomarkers and therapeutic targets, as they have been found to shape the TME by releasing TGF-ß1, a potent immunosuppressive cytokine. Studies have shown that high thrombocyte counts within the tumour would result in worse tumour control with ICIs [59]. Pharmacologically targeting thrombocyte function with Aspirin and Clopidogrel in mice, thus reducing thrombocyte presence in the tumour, resulted in better tumour control in conjunction with ICIs [60]. 

Transcriptomic analysis of all of these tumour-infiltrating cells has been proposed as a viable way to identify biomarkers, for instance, by finding gene-expression signatures that may distinguish responders from non-responders. A study by Gide et al. performed transcriptomic analysis of tumour-infiltrating T cells in tumour biopsies of melanoma patients and revealed that an effector T-cell phenotype overexpressing *EOMES*, *CD69*, and *CD45RO* is a feature of responders [61]. 

Small molecules in the TME may also be an exploitable biomarker resource. IFN-γ itself has been a target of study. This has led to the study of a potential expression signature related to IFN-γ in tumour biopsies as a biomarker of response to ICI immunotherapy. Some studies of NSCLC patients have thus identified a gene-expression signature composed of four genes (*CD274*, *CXCL9*, *IFNG*, *LAG3*) that appears to be upregulated in patients that respond to Durvalumab [62]. All of these examples show the TME biomarker potential that is yet to be explored.

## 5. Liquid Biomarkers

So far, we have described the available and potential sources of tumour-based biomarkers, which rely inherently on obtaining a sample of the primary tumour. Although these tumour-based biomarkers are better understood and more clinically validated, they are also bound by at least three important limitations: (i) the static nature of the information. Tumour biopsies are typically taken only once prior to treatment, and resampling is usually not possible; (ii) tissue sampling bias: the representativity of the sample (for instance in terms of clonal heterogeneity) is highly dependent on the operator, tumour location, and whereabouts in the tumour it is taken from and thus may be considerably biased by tissue sampling; and (iii) the procedures needed to obtain them are highly invasive and inconvenient for the patient. In the past few years, liquid biopsy strategies have emerged that focus on analysing tumour products found in many different body fluids, including blood, plasma, pleural, pericardial and cerebrospinal fluids, urine, ascites, and saliva [63,64]. These allow for easier sample retrieval as well as the potential for monitoring disease progression and treatment response over time. Some of these strategies have been studied for their potential use in finding biomarkers of response to ICIs (Table 2).

### 5.1. Ciculating Tumour DNA

Circulating free DNA (cfDNA) is genetic material released into the bloodstream or other fluids by cells. Although the exact mechanism by which this genetic material is released is not yet clear, it is believed it may be caused by freeing of the cell contents during apoptosis or necrosis but perhaps also by other unknown mechanisms. Circulating tumour DNA (ctDNA), however, is genetic material specifically of tumour origin. As with cfDNA, the exact mechanism of ctDNA release by tumour cells is not fully understood, but it includes active segregation and/or release by apoptotic or necrotic tumour cells [65]. 

The most simplistic approach to utilise ctDNA as prognostic factor in NSCLC is by quantifying its absolute levels in circulatory fluids. cfDNA can easily be extracted from blood plasma by anion exchange chromatography, with different kits employing either exchange columns or magnetic beads [66,67]. Total cfDNA concentration in blood plasma has been reported to be four times higher in the average cancer patient compared to healthy individuals, which may be valuable to identify candidate patients [68]. Other studies have revealed that advanced-stage lung cancer patients (stage III and IV) did present higher cfDNA levels than healthy controls (mean 60 ng/mL in lung cancer patients vs. mean 5 ng/mL in controls) [69]. Unfortunately, measurements of total cfDNA levels cannot be used as informative biomarker due to varying yields of cfDNA not only in different cancers but also within the same type of cancer [70].

Quantitative PCR (qPCR) analysis can then be used quantify the ctDNA fraction from total cfDNA by detecting known mutations. In NSCLC and many other tumour types, total ctDNA levels have been found to correlate with the size of tumour mass [65,71], and this may be a cost-effective way of preliminary screening for tumour presence and volume. Studies of NSCLC patients treated with anti-PD-1/PD-L1 agents have characterised and quantified total ctDNA by determining allele fraction of cancer-associated somatic mutations in blood plasma using an NGS gene-panel approach. These found that a significant drop in ctDNA levels was associated with clinical response to immunotherapy and prolonged survival [72,73]. These findings have also been validated in metastatic melanoma by digital droplet PCR (ddPCR), where quantitative monitoring of ctDNA revealed that an increase in ctDNA levels by weeks 2 or 4 of treatment was associated with no clinical benefit and eventual disease progression, while no increase in ctDNA was associated with better OS, PFS, and durable clinical benefit [74,75,76]. In NSCLC patients that have undergone primary tumour resection, ctDNA quantification showed that higher plasma ctDNA levels were associated with a worse OS, and increased ctDNA levels could be observed shortly before a patient experienced a relapse [77]. Another study by Moding et al. on NSCLC patients undergoing treatment with chemoradiotherapy (CRT) and ICIs found that the detection of ctDNA in NSCLC patients could predict response to both chemoradioterapy and ICI treatment with patients that presented no detectable ctDNA levels after either treatment presenting considerably higher PFS [78].

ctDNA may also provide valuable information of the primary tumour. For example, ctDNA sequencing can help identify actionable mutations that are sensitive to targeted therapies in the same way that can be determined from a solid biopsy sample [79]. Moreover, this method of tumour assessment can be conducted dynamically throughout treatment, thereby allowing us to monitor the changes happening to tumour clonality on a real-time scale. A study by Pérez-Barros et al. argued that samples of pleural fluid, pericardial fluid, CSF, and ascites are commonly obtained from the patient during the clinical management of NSCLC and may be of use to find actionable mutations [63]. In their study, they compared ctDNA analysis by ddPCR in these fluids vs. plasma to determine efficacy of detection of *EGFR*-sensitizing and resistance mutations. Overall, they found a higher amount of cfDNA in these other body fluids (1.90 vs. 0.36 ng/µL), and more fluid samples tested positive for *EGFR* mutations (21 vs. 16 samples). Furthermore, mutant allele frequencies (MAFs) observed in the other body fluids were higher for *EGFR*-sensitizing and resistance mutations (15.8 vs. 0.8% and 8.69 vs. 0.16%, respectively. Similar studies have been carried out using ctDNA from urine finding strong concordance in *EGFR* mutations between baseline tumour-urine samples (82% vs. 84%, respectively) [80,81].

ctDNA may also be used as a proxy to determine TMB. Sequencing ctDNA using a panel spanning NSCLC relevant genes may help determine the tumour’s mutational status. Such analysis can reveal its current TMB in peripheral blood and also denominated blood TMB (bTMB) [82]. This measure of TMB is less invasive and allows more dynamic access to the information of the tumour than conventional TMB calculation using a biopsy. A recent study by Wang et al. already employed bTMB and demonstrated it efficacy. By building a cancer gene panel encompassing 150 different cancer related genes, the NCC-GP150, they calculated bTMB levels of a cohort of 50 NSCLC patients receiving anti-PD-1/PD-L1 immunotherapy [83]. They found that a bTMB of ≥6 mut/Mb was associated with response to ICIs. 

bTMB is, however, bound by the low amount of available ctDNA, and a sufficient amount of high-quality DNA is necessary to avoid underestimation of bTMB for correct evaluation and stratification of the patient. New studies have tried to avoid this issue by turning to other more readily available fluids, such as urine. In bladder cancer patients, for example, Chauhan et al. successfully calculated TMB from urine ctDNA [84]. Translating this type of study to immunotherapy in NSCLC patients could be of interest to therefore avoid the present limitations.

Nonetheless, using ctDNA also presents with some drawbacks. The first limitation is the ability to distinguish ctDNA from cfDNA, which occurs naturally in the circulation [85]. The second limitation is that panel-based sequencing methods usually have a limit of detection (LOD) usually ranging from 0.3% to 1%, and this limits the amount of rare ctDNA molecules that can be distinguished [86]. Additionally, the ratio of cfDNA to ctDNA is variable and unknown, and this cfDNA directly affects the LOD. Conversely, ddPCR approaches, although limited to a small set of targeted mutations, can overcome the LOD limitation by being extremely sensitive, allowing absolute quantification of mutant copies per ml of blood plasma and avoiding quantification bias due to variation in non-tumour cfDNA. The extent to which this low detection is clinically actionable is, however, up to debate.

### 5.2. Ciculating Tumour Cells

CTCs (circulating tumour cells) are also widely studied as a source for biomarkers in the field of liquid biopsy. CTCs are tumour cells mainly released into the bloodstream by either the primary tumour or metastatic growths [87]. Because some tumour cells can undergo epithelial-to-mesenchymal transition (EMT), this allows them to infiltrate blood and lymphatic vessels and be actively disseminated [88]. Although the process is not yet fully understood, CTCs could provide with invaluable information about the primary tumour or any possible metastases and the dynamic changes over the course of treatment.

Unfortunately, CTCs are very difficult to obtain due to their low numbers in blood, with at most 1 in 1,000,000 cells being a CTC in most tumour types. For this reason, CTC-enrichment methods have been developed to increase their proportions within samples for a more effective detection, most of which rely on the selection of circulating cells expression epithelial surface markers, such as EpCAM and cytokeratins [89]. For example, the CellSearch technology permits the identification and quantification of individual CTCs with prognostic value in various types of cancers [90,91,92]. However, such technologies have the clear limitation that CTCs that have undergone EMT will have likely ceased to express these epithelial markers, and this might explain the poor performance in some cancers, particularly NSCLC [93,94].

Thus, there has been a tendency towards EpCam-independent enrichment technologies, including size-based enrichment mechanisms, like the Parsortix system, the vortex, or Isolation by SizE of Tumour cells (ISET) technologies [95,96,97,98,99,100]. Methods based on negative depletion of haematopoietic cells may also be appropriate since they avoid the selection of epithelial-only cells whilst removing confounding cells, such as macrophages or myeloid-derived suppressor cells (MDSCs) expressing PD-L1 [101,102].

Because CTCs are live cells, they can provide dynamic information of great use in the prediction of response to immunotherapy. PD-L1 evaluation in CTCs could be a good reflection of the PD-L1 state of the tumour without the need of a biopsy, and a correlation has been found between the two in NSCLC patients [103]. Furthermore, the detection of PD-L1-positive CTCs in advanced NSCLC patients six months after treatment with Nivolumab correlated with resistance to treatment and disease progression [104]. In another study, PD-L1-positive CTCs were detected in 100% of patients that did not respond to immunotherapy [97]. However, this presence of PD-L1+ CTCs in all non-responders reported by Guibert et al. is contradicted by many other studies that report that the presence of PD-L1+ CTCs does not predict response to ICIs [105,106]. Furthermore, concordance between PD-L1 positivity in tumour tissue and CTCs was also contradictory, with a higher percentage of PD-L1-positive cells in CTCs collected than in tissue. This indicates that the ISET method they employed may have been selecting other PD-L1+ cells that are not CTCs, such as those described above.

Another option to possibly increase the availability of CTCs is to explore other fluids. Although CTC detection techniques are mostly adapted for detection in blood, samples from pleural effusions have been found to contain CTCs in NSCLC patients. Thompson et al., for instance, analysed whether CTCs could be detected in these samples using the CellSearch system and were successful to find CTCs in 63 out of 66 patients [107]. They found more CTCs in patients with malignant pleural effusions compared to patients with benign effusions (median 1798 vs. 8 CTCs). Therefore, exploring other fluids may be of interest in NSCLC, particularly in those patients where CTCs are undetectable in the bloodstream.

Additionally, exploring other markers within CTCs could reveal new prognostic and diagnostic factors. In a study by Papadaki et al., indoleamine-2,3-dioxygenase (IDO) expression in CTCs showed promising results in predicting response to immunotherapy [105]. Detection of IDO+ CTCs was associated with shorter PFS and shorter OS. IDO expression has been shown to promote immune suppression in melanoma tumours through activation and recruitment of myeloid-derived suppressor cells [108].

High throughput sequencing of CTCs is yet another possibility, though it has been a challenge due to the limited number of cells available from each patient. Some studies have employed the use of whole genome sequencing (WGS), WES, or targeted gene panels to detect and track mutations that can help in early detection of cancer and actionable mutations for targeted therapies in many cancer types but not NSCLC [109,110].

Moreover, single-cell RNA-sequencing (scRNA-seq) also stands as a possibility for omic studies that is yet under development. Early works have showed that only 45–60% of CTCs contained RNA with enough integrity for sequencing [111,112]. Very recently, Cheng et al. used a technique called Hydro-Seq to prevent contamination by blood cells, obtaining enough CTCs with RNA of sufficient integrity [113]. This allowed them to perform scRNA-seq on CTCs of breast cancer patients, finding expression of markers of EMT and various drug targets of breast cancer. Another study performed similar transcriptomic analysis on breast cancer CTCs, finding gene-expression profiles able to identify CTCs with responsiveness specifically to oestrogen treatment [114]. These advances open up the possibility to extend CTC omic studies to NSCLC immunotherapy response in the near future.

### 5.3. Extracellular Vesicles

Extracellular vesicles (EVs) are particles delimited by a lipid bilayer that are released by many types of cells [115]. These vesicles vary in size and origin and may be microvesicles produced by exocytosis of cellular components by cell membrane budding, exosomes produced by initial endocytosis of cell components and posterior exocytosis or apoptotic bodies released during programmed cell death. They are a very important vector for intercellular communication and transportation of proteins and RNA between normal cells [116]. Cancer cells are also known to produce EVs and release them into the TME and the circulatory system. Because they are encased, these particles preserve well proteins and DNA molecules and even small RNA molecules that would otherwise be sensitive to degradation [117]. DNA fragments encased in these vesicles may span the whole genome of a cell, and so EVs released by tumours may be used to study the genome of its tumour of origin. Various studies have also shown the ability of these cancer EVs to promote immune evasion through mechanisms such as T-cell and NK cell inactivation [118]. 

EVs can be isolated from a variety of body fluids through different methods, which include separation by size (using ultrafiltration, exclusion chromatography, or microfluidic systems), by density (using differential ultracentrifugation or flotation density gradient centrifugation), and by surface-marker selection (using flow cytometry and sorting or other immunocapture techniques), and importantly, their surface and contents can be reliably analysed [117,119]. Studies have shown promising ability as prognostic and diagnostic markers in NSCLC for EVs extracted from blood plasma and pleural fluid. For example, the presence of preferentially expressed miRNAs in NSCLC has been evaluated, leading to a discovery of nine miRNAs present in pleural effusion-derived EVs in NSCLC patients but absent in pleural effusions of patients with tuberculosis or other lung lesions [120,121,122]. 

However, so far very few studies have focused on EVs as biomarkers of ICI response. The majority of the published works have focused on EVs containing PD-L1, either on the membrane or as an encased mRNA, to act as proxies for *PD-L1* overexpression in the tumour. Kim et al. performed an in-vitro experiment in which they characterised PD-L1 abundance on the membrane of EVs by immunohistochemical staining and found that plasma EVs from NSCLC patients showed PD-L1 localisation to their membrane proportional to PD-L1 positivity of the primary tumour [123]. A study by del Re et al. examined plasma derived EVs in 8 NSCLC patients and 18 melanoma patients that received Nivolumab or Pembrolizumab and performed ddPCR to measure *PD-L1* mRNA levels within these vesicles [124]. They found a significantly higher number of *PD-L1* mRNA copies in responders compared to non-responders before treatment with ICIs (mean 830.4 vs. 204.0 copies per ml, respectively). In addition to this, *PD-L1* mRNA copies significantly decreased after two months of ICI treatment in responders, whilst mRNA copy number increased in non-responders (mean 242.5 vs. 416.0 copies per ml, respectively). The study of EVs containing PD-L1 is of especial interest as a new therapeutic target in non-responders to immunotherapy. Poggio et al. used CRISPR/Cas-9 mutagenesis to knock out genes responsible for release of exosomes containing membrane PD-L1 in mouse models and found that it inhibited tumour growth [125]. They also found that PD-L1 exosomes suppressed T-cell activation, and they proposed that anti-PD-L1 antibodies functioned additively and not redundantly with exosomal PD-L1, suggesting active release of these PD-L1 exosomes directly interferes with the correct function of ICIs. Another study by Chen et al. had similar findings in metastatic melanoma patients, reporting that stimulation with IFN-γ increased PD-L1 in tumour-secreted exosomes and proposed that patients can be stratified into responders or non-responders to immunotherapy according to the increase in circulating PD-L1 during initial treatment with ICIs [126]. These studies set exosomal PD-L1 as a promising biomarker of response that needs further investigation.

### 5.4. Other Blood Markers

Other sources of biomarkers from peripheral blood samples have also been studied to track response of NSCLC patients to immunotherapy, especially cell counts. This is due to their easy and minimally invasive access and low technical assay costs due to existing clinical standardised protocols of cell detection, isolation, and counting.

#### 5.4.1. T Cells 

Peripheral cytotoxic T cells (T_c_; also described as T_CD8+_ cells) have also been investigated as treatment response biomarkers due to their easy isolation via cell-sorting techniques. In a study by Kamphorst et al. that examined NSCLC patients that received immunotherapy (Nivolumab or Pembrolizumab) as second-line treatment, an increase in cytotoxic T_CD8+PD-1+_ cell levels was found one month after treatment in 80% of responders as opposed to only 30% of non-responders that showed an increase in these cells [127]. Another study by Ottonello et al. reported significantly higher baseline T_CD8+_ cell levels in NSCLC patients that respond to immunotherapy compared non-responders [128]. However, the opposite was true in the case of T_CD8+PD-1+_ cells.

A way in which T_CD8+PD-1+_ cells have shown predictive value in response to immunotherapy is through the examination of the T-cell receptor (TCR) repertoire. The TCR sequence contains a variable section, the complementarity-determining region 3 (CDR3), which ensures antigen specificity and strength of the immune response and is the result of diversification and genetic recombination mechanisms during T-cell development, giving each T cell a unique TCR sequence [129]. Once a T cell becomes activated during the immune response, it undergoes clonal expansion, with all resulting T cells having identical TCR sequences. The collection of all different TCR sequences, their distribution, and their frequency, form the TCR repertoire of an individual. Changes in the TCR repertoire can be examined through high-throughput sequencing and have been explored in many types of cancer, including NSCLC. For instance, Han et al. focused on the TCR repertoire of T_CD8+PD-1+_ cells of NSCLC patients that were treated with anti-PD-1 or anti-PD-L1 ICIs [130]. They found that pre-treatment TCR repertoire diversity predicted response to immunotherapy, with patients with high TCR variability showing a significantly higher mean PFS than those with lower TCR variability (6.4 months vs. 2.5 months). The same was found in post-treatment samples (4–6 weeks after first treatment), with a significantly higher PFS in patients with high TCR variability compared to patients with low TCR variability (7.2 months vs. 2.6 months).

The presence of T helper cells (T_h_ or T_CD4+)_ is another potential biomarker of ICI response. In the previous study by Ottonello and colleagues, they also found significantly higher levels of T_CD4+_ cells in responders, suggesting that measuring peripheral T_CD4+_ cell levels in NSCLC patients may be helpful in predicting ICI response [128]. 

Regulatory T cells (T_regs_ or T_CD4+CD25+FOXP3+_) are in charge of maintaining immune homeostasis to prevent autoimmune reactions, therefore having immune suppressive activity. A study by Koh et al. investigated peripheral blood immune cells for their potential as biomarkers, using a discovery cohort of 83 NSCLC patients and a validation cohort of 49 NSCLC patients [131]. They found significantly higher levels of T_CD4+CD25+FOXP3+_ cells 1 week after anti-PD-1 treatment in responders.

#### 5.4.2. Immune Cell Transcriptomics

A more recent approach to find biomarkers of immunotherapy response has been to transition from pure cell counts to full omic characterisation.

Measurement of the enzyme lactate dehydrogenase (*LDH*) in peripheral blood is an example. *LDH* upregulation is linked to tumorigenesis, as it promotes the Warburg effect [132]. There is considerable evidence that high LDH levels in peripheral blood are associated with poor ICI response, specifically low PFS and OS [133]. Since peripheral blood LDH levels of patients are routinely measured, pre-treatment levels of LDH may be a predictive biomarker that can be easily implemented in clinical practice.

The cytokine IFN-γ, an important regulator of the TME, is also subject to transcriptomic analysis. As mentioned before, IFN-γ is an inflammatory cytokine release by T cells and NK cells to recruit other immune cells as part of the immune response but also mediates the upregulation of *PD-L1* in tumour cells. The group of Giunta et al. investigated *IFNG* expression in peripheral blood mononuclear cells (PBMCs) of 18 baseline blood samples of melanoma patients eligible for immunotherapy [134]. They found significantly higher levels of *IFNG* mRNA in bulk PBMCs in responders compared to non-responders. Cell-sorting analyses discovered that this increase was mostly due to higher levels of T_CD4+IFN-γ+_ and T_CD8+IFN-γ+_ cells in responders.

Transcriptomic analysis of T cells has also been explored. In 2020, Fairfax et al. performed bulk RNA-sequencing of peripheral blood T_CD8+_ cells of metastatic melanoma patients [135]. T-cell clones were identified and mapped to the samples using MiXCR to identify large clones (occupying > 0.5% of repertoire) in patients with durable response [136]. When performing single-cell RNA-sequencing of post-treatment samples, these large clones presented overexpression of certain genes involved in cytotoxicity, such as *CCL4*, *GNLY*, and *NKG7*. The identification of this expression signature three weeks after therapy onset may therefore be helpful to identify responders. Unfortunately, these signatures were not reported in treatment naïve patients, suggesting that this biomarker may only be useful once treatment has started. Although this type of transcriptomic analysis has only been carried out in melanoma patients, results are promising and could be extended to investigate other cancer types, like NSCLC.

#### 5.4.3. Soluble PD-L1

Apart from membrane bound PD-L1, PD-L1 also exists in a soluble form (sPD-L1), an isoform that lacks transmembrane domain and thus is not bound to a cell membrane. sPD-L1 can be found in blood plasma of patients with many different types of cancer, including NSCLC, and high levels of this isoform have been found to correlate with poor response to immunotherapy [137]. A study examining sPD-L1 levels in 39 NSCLC patients also found that most patients with high levels (75% of patients) did not respond to Nivolumab, while a high proportion of patients (59%) with low sPD-L1 showed durable response to Nivolumab [138]. However, analysis of sPD-L1 is still problematic. Analysis methods of sPD-L1 quantification in blood plasma vary between studies (rendering low reproducibility), and an optimal measurement method and cut-off levels linked with therapeutic response are not well defined. Moreover, since this protein is found in its soluble form, it is impossible to track its cell of origin and so may well be the case that normal cells may be releasing it, thus contributing further noise. In fact, sPD-L1 has been found to increase in blood serum of patients without cancer in processes such as inflammation, autoimmune diseases, and even during pregnancy, demonstrating low sPD-L1 specificity [139,140,141].

#### 5.4.4. Microbiome

The influence of the microbiome in the context of cancer immunoregulation has been explored in several tumour types, such as melanoma, pancreas, prostate, breast, or SCLC. However, the mechanisms by which the microbiota may play a role in regulating cancer immunotherapy are not yet clear [142]. In this regard, some works have been pursued in the field of faecal matter transplant (FMT) as a way to demonstrate the effect of the microbiome on the outcome of immunotherapy and response boosting. Derosa et al. performed FMT from responders and non-responder melanoma patients to mice and found that those transplanted from responders produced a more pronounced anti-tumour response [143].

For NSCLC, some works have analysed the levels of plasmatic citrulline (a marker of gut barrier integrity) in treatment naïve samples of these patients, with high citrulline significantly associated with clinical benefit from Nivolumab and a significantly higher PFS in citrulline-high patients [144]. Trials that attempt to determine and quantify the gut microflora and gene- and protein-expression levels have also started to launch for some cancer types [145]. The presence of DNA from the *Peptostreptoccocus*, *Paludibaculum*, or *Lewinella* families was associated to complete or partial response to Nivolumab, whereas detection of bacterial DNA of the *Gemmatimonas* family was associated with disease progression in NSCLC patients [144]. However, results have been conflicting with regards to the influence of specific microorganisms in immunotherapy response, and there is still a great deal of work that needs to be produced in order to determine what organisms are responsible for influencing ICI response [146,147,148].

## 6. Future Perspectives

In this review, we have described multiple biomarkers that are currently under development to improve the predictive stratification of NSCLC patients with regards to treatment outcome. While many of them are very promising, the current guidelines only contemplate *PD-L1* expression and TMB within the primary tumour as informative enough to guide clinical practice. Although they may provide relevant information and their use to determine ICI indication is undeniable, there is still a huge margin for improvement.

Importantly, one of the shortcomings of PD-L1 determination (and of other tumour-based biomarkers, for that matter) is the necessity of obtaining a tissue biopsy. For this reason, liquid biopsy markers, which are more easily accessible and can be determined at multiple times over the course of treatment, have started to arise as a step forward towards easier diagnosis and disease management. Liquid biopsy approaches constitute in this regard a feasible and less invasive alternative, and although samples of the approaches may be currently costly, these costs will be greatly reduced in the near future, particularly if these approaches become standard procedure. 

One of the great expectations in the field is the potential of ctDNA analysis. It is probably the most easily implementable strategy for clinical practice due to many companies developing capture-based NGS gene panels and platforms becoming more accessible over time in terms of costs and results turnaround. ctDNA analysis is an easy way to monitor tumour alterations throughout treatment, thus allowing for flexible therapeutic decision-making that matches tumour evolution. Another possibility that this approach using ctDNA offers is the chance to indirectly assess TMB. Although this possibility has not yet been validated, current research indicates that with appropriate panel design it would be possible to determine both specific NSCLC alterations and TMB. 

Because ctDNA particles are somewhat limited and may actually be the product of apoptotic tumour cells, CTC analysis arises as an alternative to evaluate tumour cells directly, with a particular emphasis on stopping the metastatic spread. However, the potential gain in informativeness comes at a price, with identification and isolation technologies lagging behind those of ctDNA. This proves especially problematic in NSCLC, where the number of found CTCs in blood have been consistently smaller than in other cancer types. This may be at least partly because CTC isolation technologies rely largely on EpCAM-dependent technologies that are only capable of identifying epithelial cells and cannot retain cells that have undergone EMT. Moreover, the genetic material from CTCs has recently started to be reliably sequenced and will provide invaluable information about both the primary tumour and the metastases. This could allow the examination of the transcriptome and mutational status of the cells most relevant to secondary growths at different time points, with the potential to stop the spreading in real time.

Other alternative strategies have also gained traction in the panorama of immunotherapy monitoring. Analyses of EVs have shown potential as a less invasive substitute of conventional *PD-L1* expression assaying of the primary tumour, since both membrane-bound PD-L1 and mRNA can be found in these vesicles. Since EVs can be derived from many types of cells and not only cancer cells, specific markers must be found to reliably distinguish those of tumour origin. However, the wide variety of sources and types of EVs make for a promising area of study to find novel biomarkers. T-cells counts and TCR determinations and gene-expression quantifications also show great promise in the response biomarker setting, with possibilities ranging from single-gene quantifications, like that of *IFNG*, to gene molecular signatures, to full translational patterns depicting the most relevant cellular processes that play a role in treatment response.

Much work still lies ahead. Currently, most of these liquid biopsy approaches are at an on-going research phase, and once the first indications of the potential relevance of these markers have been gathered, it is essential to carry out preliminary studies and clinical trials to evaluate the reproducibility and applicability of these technologies towards a future implementation in clinical practice. There are still several important issues that need to be considered for response biomarker analyses, such as how clonal variability affects treatment response and biomarker development and how disease molecular heterogeneity affects biomarker development towards the standardization of protocols. Moreover, the search for biomarkers has been extensively focused on blood plasma, and as we have already seen, other bodily fluids may be sometimes more useful for NSCLC and thus should be explored to their full potential. Furthermore, it is possibly incredibly naïve to think that a single biomarker may be the solution to predicting treatment response accurately enough to guide clinical management. Instead, multiparametric scores combining several biomarkers may provide a more robust approach for a more precise prediction of ICI response.

In summary, liquid biopsy approaches offer an unprecedented possibility to explore biomarkers of immunotherapy response via less invisible and potentially more cost-effective methods. Several strategies, such as ctDNA analyses, have already provided relevant data with a clear clinical applicability, with others lagging not far behind but relying on current technological developments. These approaches will surely grant us the possibility in the near future to better ascertain the individuals most likely to benefit from ICI treatment, thereby allowing for tailored treatment and personalised medicine approaches.

## Figures and Tables

**Figure 1 jpm-11-00971-f001:**
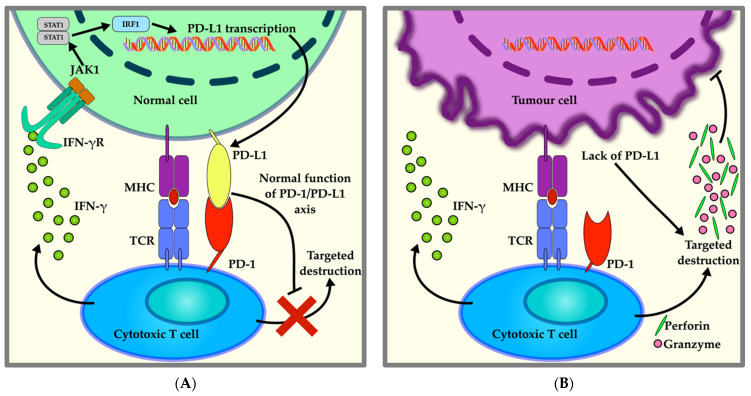
Immune targeting of cells by cytotoxic T cells: (**A**) Mechanism of immune tolerance of a native cell by the activation of the PD-1/PD-L1 axis. A T cell is activated by an antigen presented on a major histocompatibility complex (MHC) molecule on the surface of the cell. This triggers the release of inflammatory cytokines, such as interferon gamma (IFN-γ), that bind its receptor (IFN-γR) on the cell membrane, upregulating the expression of *PD-L1*. The interaction between PD-L1 and PD-1 on the T-cell surface works as a mechanism of cell recognition and promotes immune tolerance, preventing the destruction of the cell. (**B**) Mechanism of immune targeting of a tumour cell. Conversely, when the cytotoxic T cell recognizes the antigen presented on an MHC, but no PD-L1 is expressed, the release of IFN-γ and cytolytic mediators (perforin and granzymes) results in (tumour) cell elimination. (**C**) Mechanism of immune evasion of the tumour cell by exploiting PD-1/PD-L1 axis. The cytotoxic T-cell activation recognizes the tumour antigen presented on its MHC, but PD-L1 transcription is activated by IFN-γ. PD-L1 inhibits anti-tumour activity of the cytotoxic T cell. (**D**) Mechanism of PD-1/PD-L1 blockade by anti-PD-1 or anti-PD-L1 immune checkpoint inhibitors (ICIs). The antibodies block the interaction between PD-1 and PD-L1, preventing inhibition of cytotoxic T-cell antitumor activity and triggering the release of perforin and granzymes for the elimination of the tumour cell and thereby surpassing the PD-1/PD-L1 blockade.

**Table 1 jpm-11-00971-t001:** List of anti-PD-1/PD-L1 immune checkpoints inhibitors that have been approved by different government agencies.

Drug	Indication ^1^
Nivolumab (Bristol-Myers Squibb, New York, NY, USA) Anti-PD-1	Non-Small Cell Lung Cancer (FDA, EMA, PMDA, NMPA) Melanoma (FDA, EMA, PMDA) Renal Cell Carcinoma (FDA, EMA, PMDA) Head and Neck Cancer (FDA, EMA, PMDA) Hodgkin Lymphoma (FDA, EMA, PMDA) Oesophageal Cancer (FDA, EMA, PMDA) Bladder Cancer (FDA, EMA) Gastric Cancer (FDA, PMDA) Mesothelioma (PMDA) Colorectal Cancer (FDA) Hepatocellular Cancer (FDA) Small Cell Lung Cancer (FDA)
Pembrolizumab (Merck Co., Kenilworth, NJ, USA) Anti-PD-1	Melanoma (FDA, EMA, PMDA, NMPA) Non-Small Cell Lung Cancer (FDA, EMA, PMDA, NMPA) Renal Cell Carcinoma (FDA, EMA, PMDA) Hodgkin Lymphoma (FDA, EMA, PMDA) Bladder Cancer (FDA, EMA, PMDA) Head and Neck Cancer (FDA, EMA, PMDA) MSI-High Solid Tumours (FDA, PMDA) Merkel Cell Carcinoma (FDA, EMA, PMDA) Oesophageal Cancer (FDA, EMA, PMDA) Gastric Cancer (FDA) Hepatocellular Carcinoma (FDA) Cervical Cancer (FDA) Primary Mediastinal B-cell Lymphoma (FDA) Small Cell Lung Cancer (FDA) Endometrial Carcinoma (FDA) Cutaneous Squamous Cell Carcinoma (FDA) Triple Negative Breast Cancer (FDA)
Atezolizumab (Roche, Basel, Switzerland) Anti-PD-L1	Non-Small Cell Lung Cancer (FDA, EMA, PMDA) Small Cell Lung Cancer (FDA, PMDA) Bladder Cancer (FDA, EMA) Breast Cancer (FDA) Hepatocellular Carcinoma (FDA) Melanoma (FDA)
Durvalumab (AstraZeneca, Cambridge, UK) Anti-PD-L1	Non-Small Cell Lung Cancer (FDA, EMA, PMDA) Bladder Cancer (FDA) Small Cell Lung Cancer (FDA)
Avelumab (Pfizer/Merck KGaA, Darmstadt, Germany) Anti-PD-L1	Merkel Cell Carcinoma (FDA, EMA, PMDA) Renal Cell Carcinoma (FDA) Bladder Cancer (FDA)
Cemiplimab (Regeneron, New York, NY, USA) Anti-PD-L1	Cutaneous Squamous-Cell Cancer (FDA, EMA) Non-Small Cell Lung Cancer (FDA) Basal Cell Carcinoma (FDA)
Toripalimab (Junshi Biosciences, Shanghai, China) Anti-PD-1	Melanoma (NMPA) Nasopharyngeal Carcinoma (NMPA)
Sintilimab (Innovent Biologics, Hongkong, China) Anti-PD-1	Hodgkin Lymphoma (NMPA) Non-Small Cell Lung Cancer (NMPA)
Camrelizumab (Jiangsu HengRui, Lianyungang, China) Anti-PD-1	Hodgkin Lymphoma (NMPA) Hepatocellular Carcinoma (NMPA)
Tislelizumab (Beigene, Beijing, China) Anti-PD-1	Hodgkin Lymphoma (NMPA) Bladder Cancer (NMPA)
Dostarlimab (GlaxoSmithKline LLC, Wales, UK) Anti-PD-1	Endometrial Carcinoma (FDA)

^1^ Indication for each type of tumour approved by the Food and Drugs Administration (FDA), USA; the European Medicines Agency (EMA), EU; the Pharmaceuticals and Medical Devices Agency (PMDA), Japan; and the National Medical Products Administration (NMPA), China. Data adapted from Cancer Research Institute (cancerresearch.org, last accessed 30 June 2021) [23].

**Table 2 jpm-11-00971-t002:** Advantages and disadvantages of the liquid biomarkers with potential use in immunotherapy response monitoring.

Liquid Biomarker	Advantages	Disadvantages
ctDNA ^1^	-Ready access to genetic material of the primary tumour -Dynamic information over the course of treatment -Predictive value in quantification of absolute levels and alternate source for TMB calculation	-Limits of detection: allele fraction and variant calling pipelines may produce many false negatives and/or false positives -Lack of standardization of thresholds
CTCs ^2^	-Provide reflection of tumour status and tumour heterogeneity -Dynamic information over the course of treatment -Predictive value in quantification of absolute cell counts, examination of cell membrane markers expressed, and omic characterisation	-Disparity in methods used for isolation and enrichment -Identification and isolation methods require high sensitivity -False negatives -Limited and fragile population -Low yield of genetic material
EVs ^3^	-Many different types available from different sources -Stable, can efficiently preserve contents -Can provide protein and lipidic markers, and genetic material	-Contained genetic material is very limited -High heterogeneity makes it difficult to distinguish EVs of tumoral origin
T_c_ counts ^4^	-Standard, reliable methods of isolation. -Easy identification and count through flow cytometry -Dynamic information over the course of treatment	-Lack of standardization of thresholds -Levels greatly vary from patient to patient -Contradictions when considering marker PD-1
T_h_ counts ^5^	-Standard, reliable methods of isolation. -Easy identification and count through flow cytometry -Dynamic information over the course of treatment	-Lack of standardization of thresholds -Levels greatly vary from patient to patient
T_regs_ counts ^6^	-Standard, reliable methods of isolation. -Easy identification and count through flow cytometry	-Lack of standardization of thresholds -Levels greatly vary from patient to patient -Requires multiple markers for high sensitivity of detection
TCR determination ^7^	-Segment of DNA of interest is short, well defined, and has been explored in detail -Multiple protocols and kits available to amplify region of interest.	-Requires analysis of many individual cells to be significant -Careful selection of specific population of interest is required
Transcriptomic analysis of T cells	-High yield of genetic material from the high amount of T cells available in peripheral blood. -Dynamic information over the course of treatment -Activation of different pathways may be tracked throughout treatment.	-Lack of reproducibility—multitude of expression signatures that vary between experiments -Disparity in methods used to identify significant expression signatures
IFN-γ expression ^8^	-Specific, well-defined marker -High availability -Standardised, targeted methods of quantification	-Expressed by many cell types. Not specific to response. Cells of interest must be selected
sPD-L1 ^9^	-Great focus in immunotherapy -Specific, easily targetable marker -Direct, less invasive alternative to *PD-L1* expression in primary tumour	-Lack of standardization of thresholds -Has same accuracy limitations as conventional PD-L1 assay
LDH ^10^	-Cheap, easy method of enzyme quantification	-Lack of standardization of thresholds
Microbiome	-Many novel potential markers available to study	-Still in very early stages of study -Microbiota that determine response to immunotherapy widely vary between different types of cancer -Lack of standardization, no fixed markers across studies

^1^ Circulating tumour DNA. ^2^ Circulating tumour cells. ^3^ Extracellular vesicles. ^4^ T-cell counts. ^5^ T helper-cell counts. ^6^ T regulatory cell counts. ^7^ T-cell receptor repertoire. ^8^ Interferon gamma expression. ^9^ Soluble PD-L1. ^10^ Lactate dehydrogenase.

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
