# Peer review of "Liquid Biopsy Biomarkers for Immunotherapy in Non-Small Cell Lung Carcinoma: Lessons Learned and the Road Ahead"

_jpm, 2021, doi:10.3390/jpm11100971_

Round 1

Reviewer 1 Report

Jesus Hita-Millan et al in this review summarised the recent result in the field of immune checkpoint inhibitors (ICIs) response biomarkers in non-small cell lung cancer (NSCLC) patients and highlights advantages and limitations. Implementation of ICIs has resulted in increased survival rates in NSCLC patients. However, a significant proportion of patients does not seem to respond to immunotherapy, and some individuals even develop secondary resistance to treatments. The authors discussed liquid cancer biomarkers for potential ICI response monitoring, including ctDNA, CTC, Exosomes, Tc counts, Th counts, Treg counts, et al. The authors did not distinguish the potential differences and the sources of these biomarkers such as blood, urine, or saliva.  Fig 1 is nicely presented. The authors should provide a more complex picture shown the TME and ICIs with the indication of potential liquid biomarkers.

Author Response

Response to Reviewer 1 Comments

Jesus Hita-Millan et al in this review summarised the recent result in the field of immune checkpoint inhibitors (ICIs) response biomarkers in non-small cell lung cancer (NSCLC) patients and highlights advantages and limitations. Implementation of ICIs has resulted in increased survival rates in NSCLC patients. However, a significant proportion of patients does not seem to respond to immunotherapy, and some individuals even develop secondary resistance to treatments. The authors discussed liquid cancer biomarkers for potential ICI response monitoring, including ctDNA, CTC, Exosomes, Tc counts, Th counts, Treg counts, et al.

  • The authors did not distinguish the potential differences and the sources of these biomarkers such as blood, urine, or saliva. 

We thank the reviewer for the observation that the different sources of liquid biopsy biomarkers were not appropriately highlighted. We have now revised the manuscript and included appropriate examples and references in Section 5 pages 8, 12, 13, 14.

We made sure to include most fluid sources of liquid biomarkers in section 5, and point out their advantages and disadvantages when used for ctDNA analysis, CTC detection, and extracellular vesicles analysis in their respective sections. A further comment has been added in future perspectives highlighting the importance of exploring many different types of fluids that may help find new biomarkers.

  • Fig 1 is nicely presented.

We thank the reviewer for this comment. We do think it is a complicated process and the figure helps clarify it in a visual way.

  • The authors should provide a more complex picture shown the TME and ICIs with the indication of potential liquid biomarkers.

Section 4.3 has been expanded to further discuss the importance of the TME in the context of immunotherapy, and the different classifications of the TME according to TIL presence in the tumour and PD-L1 expression. Furthermore, a section has been added discussing thrombocytes as another biomarker and therapeutic target that may modulate the TME. We hope the reviewer finds these changes appropriate.

Reviewer 2 Report

This review discusses issues related to the prediction of efficacy of PD1/PD-L1 immunotherapies, and the potential contribution of liquid biopsies. The manuscript is well structured and reports many relevant results on a wide range of biomarkers.

First, I strongly suggest that authors have their manuscript spell-checked by a native English speaker.

Some of the data seem incomplete or incorrect. Here are some suggestions for a more comprehensive treatment of the subject:

- Line 39: please replace "mutations" with "alterations" (ROS1 and ALK alterations are gene rearrangements, not mutations). In addition, other therapeutic targets can be mentioned (MET, possibly ERBB2 and KRAS p.G12C).

- Line 58: Neither PD1 nor PD-L1 target antigen for immune destruction. Please rephrase.

- Lines 88-89: "The interaction [...] triggers works as a mechanism [...]" One of the words ("triggers" or "works") should be deleted.

- Lines 99-102: In my opinion, this sentence has little to do with immune checkpoint inhibitors. I suggest deleting it to refer only to PD1 and PD-L1, consistent with the previous section and the rest of the manuscript.

- Lines 113-114: CheckMate 017 trial showed an improvement in OS (assessed by hazard ratio). The median OS is only a descriptive parameter. I suggest rewording as follows: "Moreover, CheckMate 017 showed an improved OS in NSCLC patients treated with Nivolumab compared to standard treatment with docetaxel (median OS: 9.2 months vs. 6.0 months respectively)".

- Line 133: ORR is a rate (as the name suggests). Reword: either "Patients had an ORR of 30.2% in first line" or "30.2% of first-line treated patients had an OR (objective response)". The same remark can be made on lines 134 and 173.

- Line 156: Please specify "Immunohistochemical staining”

- Section 4.2

-> Line 180 : TMB refers to the density of somatic mutations in the tumour genome (usually expressed as mutations/Mb) rather than the number.

-> I agree with the limitations stated by the authors, however I think it should be made clear that TMB is, like PD-L1 tumour expression, a very imperfect marker: a low TMB does not exclude the possibility of a response to immunotherapies, and a high TMB does not guarantee it. I also suggest to provide some efficacy data of anti-PD1 immunotherapies according to TMB in NSCLC patients to highlight this point.

- Line 216: "endosomes" seems to be incorrect given the context. Endothelial cells?

- Section 5.1

-> Line 275: cfDNA is not extracted by ultra-centrifugation, it is extracted by anion exchange chromatography, either on column or on magnetic beads, in almost all published work. Also, RT-PCR refers to Reverse Transcriptase PCR, used for RNA analysis. Please refer to real-time PCR as qPCR (for quantitative PCR).

-> The detection of ctDNA at diagnosis, especially in high amounts, is indeed an independent prognostic factor in many cancers, including NSCLC. This should be clearly distinguished from the use of ctDNA for minimal residual disease (MRD) assessment. The study by Moding et al. refers to the latter use: ctDNA is used to investigate the persistence of tumour tissue after treatment, in order to assess the risk of relapse. Detection of ctDNA after CRT is predictive of an increased risk of relapse during maintenance immunotherapy. In this context, ctDNA is not, in my opinion, a predictive biomarker of the efficacy of the immunotherapy itself.

-> On the other hand, ctDNA is indeed quantitatively correlated with tumour burden. It can be monitored during the follow-up of NSCLC patients treated with immunotherapy, to early assess the therapeutic response. A few studies have evaluated this use of ctDNA in NSCLC, for example:

  • Anagnostou V, et al. Dynamics of Tumor and Immune Responses during Immune Checkpoint Blockade in Non-Small Cell Lung Cancer. Cancer Res. 15 2019;79(6):1214‑
  • Li L, et al. Serial ultra-deep sequencing of circulating tumor DNA reveals the clonal evolution in non-small cell lung cancer patients treated with anti-PD1 immunotherapy. Cancer Med. 2019;8(18):7669‑
  • Guibert N, et al. Targeted sequencing of plasma cell-free DNA to predict response to PD1 inhibitors in advanced non-small cell lung cancer. Lung Cancer. 1 nov 2019;137:1‑
  • Goldberg SB, et al. Early Assessment of Lung Cancer Immunotherapy Response via Circulating Tumor DNA. Clin Cancer Res Off J Am Assoc Cancer Res. 15 avr 2018;24(8):1872‑

-> This use of ctDNA for monitoring purposes has also been further evaluated in metastatic cutaneous melanoma, and some authors have proposed criteria for the interpretation of ctDNA kinetics in patients treated with immunotherapy:

  • Lee JH, et al. Circulating tumour DNA predicts response to anti-PD1 antibodies in metastatic melanoma. Ann Oncol Off J Eur Soc Med Oncol. 1 mai 2017;28(5):1130‑
  • Herbreteau G, et al. Quantitative monitoring of circulating tumor DNA predicts response of cutaneous metastatic melanoma to anti-PD1 immunotherapy. Oncotarget. 2018 May 18;9(38):25265-25276.
  • Herbreteau G, et al. Circulating Tumor DNA Early Kinetics Predict Response of Metastatic Melanoma to Anti-PD1 Immunotherapy: Validation Study. Cancers (Basel). 2021 Apr 11;13(8):1826.

-> NGS is the only analytical method described in the manuscript for the analysis and quantification of ctDNA. However, other methods are widely used, in particular digital PCR (dPCR). This method overcomes some of the limitations of NGS: it allows the detection of mutated copies independently of the "background" of non-tumour cfDNA, and it also allows their absolute quantification (in mutated DNA copies/mL of plasma), thus avoiding quantification bias due to quantitative variations in non-tumour cfDNA. I believe that this method should be discussed in the manuscript. Furthermore, these analytical methods are not very complex, and I suggest to modify Table 2 and section 6 (Future perspectives) accordingly.

-> Total cfDNA is not a sufficiently reliable diagnostic marker to be use in clinical practice. See meta-analysis :

  • van der Vaart M, Pretorius PJ. Is the role of circulating DNA as a biomarker of cancer being prematurely overrated? Clin Biochem. 1 janv 2010;43(1):26‑

-> One of the main limitations of bTMB is that it requires a sufficient amount of high quality ctDNA to be evaluable, otherwise it may be underestimated and lead to poor patient stratification.

- Section 5.2

-> Classification of CTCs based on morphology and not based on the presence of EpCAM has other limitations. For example, circulating macrophages expressing PD-L1 can be misinterpreted as PD-L1 positive CTCs, as can some circulating myeloid-derived suppressive cells.

-> The results of the Guibert et al. study are in contradiction with other reported results. Please highlight and discuss the discrepancy.

- Section 5.3

-> Line 376: Several types of extracellular vesicles are described: exosomes, but also microvesicles (including large oncosomes, specifically secreted by tumour cells) and apoptotic bodies. These different subpopulations are secreted by different mechanisms, and also differ in content. I suggest either to mention these different subpopulations or to treat them globally with the established name: "extracellular vesicles".

-> These different vesicles also carry DNA fragments. These fragments cover the whole genome and can therefore be used to study the tumour genome. I suggest to specify this.

-> There are actually a wide variety of methods to isolate extracellular vesicles, either according to their density (differential ultracentrifugation, flotation density gradient), their size (ultrafiltration, steric exclusion chromatography, microfluidic systems) or the antigens present on their surface (flow cytometry, immunocapture). The different subpopulations obtained can vary greatly depending on the isolation methods. I suggest to clarify this, and to modify Table 2 accordingly.

-> I suggest to cite these two studies dealing with the role of exosomal PD-L1 and its interest as a predictive biomarker of response to immunotherapies:

  • Chen G, et al. Exosomal PD-L1 contributes to immunosuppression and is associated with anti-PD-1 response. Nature 2018; 560(7718): 382–386. 130.
  • Poggio M, et al. Suppression of exosomal PD-L1 induces systemic anti-tumor immunity and memory. Cell 2019; 177(2): 414–427.e13.

- Section 5.4.3: I suggest discussing the limitations of using sPD-L1: the optimal analytical method and the cut-off associated with a good therapeutic response are not defined. Furthermore, this biomarker is not specific, and may increase with inflammation, allergy, autoimmune diseases, infections, diabetes, age and pregnancy.

- Section 6: In the absence of an ideal biomarker for predicting response to immunotherapy, the development of multiparametric scores combining the results of several biomarkers seems an interesting perspective that I suggest to develop.

Author Response

Response to Reviewer 2 Comments

This review discusses issues related to the prediction of efficacy of PD1/PD-L1 immunotherapies, and the potential contribution of liquid biopsies. The manuscript is well structured and reports many relevant results on a wide range of biomarkers.

  • First, I strongly suggest that authors have their manuscript spell-checked by a native English speaker.

We thank the reviewer about the thorough improvements suggested for the manuscript. We have now tried to implement them in the appropriate sections (see below). With regards to the language, the corresponding author is a native English speaker. Nevertheless, we have also reviewed the language, as suggested, for any inconsistencies, and changed the spelling to British English throughout the manuscript.

Some of the data seem incomplete or incorrect. Here are some suggestions for a more comprehensive treatment of the subject:

  • Line 39: please replace "mutations" with "alterations" (ROS1and ALK alterations are gene rearrangements, not mutations). In addition, other therapeutic targets can be mentioned (MET, possibly ERBB2 and KRASG12C).

Gene rearrangements may be considered as mutations, just not point mutations. However, we agree with the reviewer that the term “mutations” is perhaps not the most appropriate, and it has now been widely deprecated towards “variation” instead. Hence, we have replaced “Mutations” with “alterations”. MET has also been included as harbouring actionable variation. Other suggested therapeutic targets (ERBB2 and KRAS) were not included however, since variation in these genes is clinically actionable in that it is indicative of resistance to TKIs, but does not constitute targeted therapy, as mentioned in the manuscript.

  • Line 58: Neither PD1 nor PD-L1 target antigen for immune destruction. Please rephrase.

We have now rephrased the sentence to “The PD-L1/PD-1 axis is a self-tolerance mechanism that protects cells from being targeted for immune destruction during an inflammatory response (line 68).

  • Lines 88-89: "The interaction [...] triggers works as a mechanism [...]" One of the words ("triggers" or "works") should be deleted.

“Triggers” has now been deleted (line 106).

  • Lines 99-102: In my opinion, this sentence has little to do with immune checkpoint inhibitors. I suggest deleting it to refer only to PD1 and PD-L1, consistent with the previous section and the rest of the manuscript.

We accept the reviewer´s comment and have now deleted this sentence.

  • Lines 113-114: CheckMate 017 trial showed an improvement in OS (assessed by hazard ratio). The median OS is only a descriptive parameter. I suggest rewording as follows: "Moreover, CheckMate 017 showed an improved OS in NSCLC patients treated with Nivolumab compared to standard treatment with docetaxel (median OS: 9.2 months vs. 6.0 months respectively)".

These lines were rephrased as suggested (lines 131-133).

  • Line 133: ORR is a rate (as the name suggests). Reword: either "Patients had an ORR of 30.2% in first line" or "30.2% of first-line treated patients had an OR (objective response)". The same remark can be made on lines 134 and 173.

The suggested change has been implemented on lines 176, 177 & 215.

  • Line 156: Please specify "Immunohistochemical staining”

“Immunohistochemical” staining was added (line 197).

- Line 180: TMB refers to the density of somatic mutations in the tumour genome (usually expressed as mutations/Mb) rather than the number.

Definition of TMB corrected, as suggested (lines 222-223).

  • I agree with the limitations stated by the authors, however I think it should be made clear that TMB is, like PD-L1 tumour expression, a very imperfect marker: a low TMB does not exclude the possibility of a response to immunotherapies, and a high TMB does not guarantee it. I also suggest to provide some efficacy data of anti-PD1 immunotherapies according to TMB in NSCLC patients to highlight this point.

We are thankful for this observation and have accordingly included it in the text, as it does make a significant point (lines 265-273).

  • Line 216: "endosomes" seems to be incorrect given the context. Endothelial cells?

Indeed, it was endothelial cells (now corrected on line 286).

  • Line 275: cfDNA is not extracted by ultra-centrifugation, it is extracted by anion exchange chromatography, either on column or on magnetic beads, in almost all published work. Also, RT-PCR refers to Reverse Transcriptase PCR, used for RNA analysis. Please refer to real-time PCR as qPCR (for quantitative PCR).

We appreciate these points and have added the appropriate descriptions on page 11.

  • The detection of ctDNA at diagnosis, especially in high amounts, is indeed an independent prognostic factor in many cancers, including NSCLC. This should be clearly distinguished from the use of ctDNA for minimal residual disease (MRD) assessment. The study by Moding et al. refers to the latter use: ctDNA is used to investigate the persistence of tumour tissue after treatment, in order to assess the risk of relapse. Detection of ctDNA after CRT is predictive of an increased risk of relapse during maintenance immunotherapy. In this context, ctDNA is not, in my opinion, a predictive biomarker of the efficacy of the immunotherapy itself.

We thank the reviewer for pointing this out. We agree that they are two different endpoints, and although we believe that MRD could be used to indirectly assess treatment efficacy, this is not an evaluation of ICI response, and have removed that reference accordingly.

  • On the other hand, ctDNA is indeed quantitatively correlated with tumour burden. It can be monitored during the follow-up of NSCLC patients treated with immunotherapy, to early assess the therapeutic response. A few studies have evaluated this use of ctDNA in NSCLC, for example:
  • Anagnostou V, et al. Dynamics of Tumor and Immune Responses during Immune Checkpoint Blockade in Non-Small Cell Lung Cancer. Cancer Res. 15 2019;79(6):1214‑
  • Li L, et al. Serial ultra-deep sequencing of circulating tumor DNA reveals the clonal evolution in non-small cell lung cancer patients treated with anti-PD1 immunotherapy. Cancer Med. 2019;8(18):7669‑
  • Guibert N, et al. Targeted sequencing of plasma cell-free DNA to predict response to PD1 inhibitors in advanced non-small cell lung cancer. Lung Cancer. 1 nov 2019;137:1‑
  • Goldberg SB, et al. Early Assessment of Lung Cancer Immunotherapy Response via Circulating Tumor DNA. Clin Cancer Res Off J Am Assoc Cancer Res. 15 avr 2018;24(8):1872‑

This finding has been detailed using two of these very useful sources that the reviewer has suggested (lines 642-661).

  • This use of ctDNA for monitoring purposes has also been further evaluated in metastatic cutaneous melanoma, and some authors have proposed criteria for the interpretation of ctDNA kinetics in patients treated with immunotherapy:
  • Lee JH, et al. Circulating tumour DNA predicts response to anti-PD1 antibodies in metastatic melanoma. Ann Oncol Off J Eur Soc Med Oncol. 1 mai 2017;28(5):1130‑
  • Herbreteau G, et al. Quantitative monitoring of circulating tumor DNA predicts response of cutaneous metastatic melanoma to anti-PD1 immunotherapy. Oncotarget. 2018 May 18;9(38):25265-25276.
  • Herbreteau G, et al. Circulating Tumor DNA Early Kinetics Predict Response of Metastatic Melanoma to Anti-PD1 Immunotherapy: Validation Study. Cancers (Basel). 2021 Apr 11;13(8):1826.

The monitoring perspective of ctDNA use is now represented on lines 661-665.

  • NGS is the only analytical method described in the manuscript for the analysis and quantification of ctDNA. However, other methods are widely used, in particular digital PCR (dPCR). This method overcomes some of the limitations of NGS: it allows the detection of mutated copies independently of the "background" of non-tumour cfDNA, and it also allows their absolute quantification (in mutated DNA copies/mL of plasma), thus avoiding quantification bias due to quantitative variations in non-tumour cfDNA. I believe that this method should be discussed in the manuscript. Furthermore, these analytical methods are not very complex, and I suggest to modify Table 2 and section 6 (Future perspectives) accordingly.

We agree with the reviewer that ddPCR and NGS are both valid approaches in ctDNA analyses. ddPCR is mainly oriented towards the identification of previously defined mutations, and is more appropriate for disease monitoring and evolution, whereas NGS provides a more agnostic approach towards tumour variant identification. Both have now been discussed in the manuscript (line 712 and Table 2).

  • Total cfDNA is not a sufficiently reliable diagnostic marker to be use in clinical practice. See meta-analysis :
  • van der Vaart M, Pretorius PJ. Is the role of circulating DNA as a biomarker of cancer being prematurely overrated? Clin Biochem. 1 janv 2010;43(1):26‑

This limitation has now been added when discussing total cfDNA (lines 634-636). It adds onto the point that not a single one of the mentioned markers has yet provided a good enough biomarker of ICI treatment response.

  • One of the main limitations of bTMB is that it requires a sufficient amount of high quality ctDNA to be evaluable, otherwise it may be underestimated and lead to poor patient stratification.

This limitation of bTMB has also been added (line 700).

  • Classification of CTCs based on morphology and not based on the presence of EpCAM has other limitations. For example, circulating macrophages expressing PD-L1 can be misinterpreted as PD-L1 positive CTCs, as can some circulating myeloid-derived suppressive cells.

We agree with the reviewer that this is an interesting issue to raise, and agree that morphological assessment is perhaps the least appropriate. Other EpCAM-independent approaches, such as negative depletion are then more suitable and allow avoidance of epithelia-only selection whilst removing blood cells (line 998).

  • The results of the Guibert et al. study are in contradiction with other reported results. Please highlight and discuss the discrepancy.

This discrepancy has now been pointed out, mentioning limitations of said study and citing other relevant works (line 1009).

  • Line 376: Several types of extracellular vesicles are described: exosomes, but also microvesicles (including large oncosomes, specifically secreted by tumour cells) and apoptotic bodies. These different subpopulations are secreted by different mechanisms, and also differ in content. I suggest either to mention these different subpopulations or to treat them globally with the established name: "extracellular vesicles".

We agree with the reviewer that the term “Exosomes” was inappropriate and didn’t correspond to the range of vesicles with use as a biomarker source. We have now changed this throughout the manuscript and have discussed different types of vesicles (Section 5.3).

  • These different vesicles also carry DNA fragments. These fragments cover the whole genome and can therefore be used to study the tumour genome. I suggest to specify this.

This is a remarkable feature that has now been added to page 14 (line 1130).

  • There are actually a wide variety of methods to isolate extracellular vesicles, either according to their density (differential ultracentrifugation, flotation density gradient), their size (ultrafiltration, steric exclusion chromatography, microfluidic systems) or the antigens present on their surface (flow cytometry, immunocapture). The different subpopulations obtained can vary greatly depending on the isolation methods. I suggest to clarify this, and to modify Table 2 accordingly.

We thank the reviewer for this extensive description, that we have now included on page 14 (lines 1135-1138) and Table 2.

  • I suggest to cite these two studies dealing with the role of exosomal PD-L1 and its interest as a predictive biomarker of response to immunotherapies:
  • Chen G, et al. Exosomal PD-L1 contributes to immunosuppression and is associated with anti-PD-1 response. Nature 2018; 560(7718): 382–386. 130.
  • Poggio M, et al. Suppression of exosomal PD-L1 induces systemic anti-tumor immunity and memory. Cell 2019; 177(2): 414–427.e13.

We thank the reviewer for these two very interesting works that have now been included in the manuscript (lines 1217-1229).

  • Section 5.4.3: I suggest discussing the limitations of using sPD-L1: the optimal analytical method and the cut-off associated with a good therapeutic response are not defined. Furthermore, this biomarker is not specific, and may increase with inflammation, allergy, autoimmune diseases, infections, diabetes, age and pregnancy.

These are very good points indeed and we have now raised them in the manuscript (lines 1335-1342).

  • Section 6: In the absence of an ideal biomarker for predicting response to immunotherapy, the development of multiparametric scores combining the results of several biomarkers seems an interesting perspective that I suggest to develop.

We very much agree with the reviewer in the observation that trying to identify a single biomarker of response is inherently naïve, and multi-marker response models are likely the way to go for further biomarker development. This has been added to future directions discussion (line 1439).

Round 2

Reviewer 2 Report

I would like to thank the authors for taking into account the majority of my comments (of which there were many, I apologise). This review is well structured and well documented. The value of liquid biopsies in predicting/detecting NSCLC response to anti-PD1/PD-L1 immunotherapies, their limitations and technical issues now seem to me to be comprehensively addressed.

Two minor comments:

  • « Other suggested therapeutic targets (ERBB2 and KRAS) were not included however, since variation in these genes is clinically actionable in that it is indicative of resistance to TKIs, but does not constitute targeted therapy, as mentioned in the manuscript.”

I accept that KRAS and ERBB2 are not mentioned, however, for the record, the KRAS p.G12C mutation is a therapeutic target, not just a TKI resistance factor: sotorasib (AMG510), a KRAS p.G12C inhibitor, is FDA-approved in NSCLC (and EMA approval is pending) based on the results of the CodeBreak 100 phase II trial. There are also other FDA-approved therapies in NSCLC, such as those targeting ERBB2 exon 20 insertions (Trastuzumab Deruxtecan), or RET gene fusions (pralsetinib, selpercatinib)

  • Line 186: " Mb" instead of "MB”

Author Response

"Please see attachment"
